# SARS-CoV-2 Omicron subvariant genomic variation associations with immune evasion in Northern California: A retrospective cohort study

**Joshua R. Nugent**[1]*, **Mariah S. Wood**[1], **Liyan Liu**[1], **Teal Bullick**[2], **Jeffrey M. Schapiro**[3], **Phacharee Arunleung**[2], **Gautham Gautham**[2], **Shiffen Getabecha**[2], **Christina Morales**[2], **Laura B. Amsden**[1], **Crystal A. Hsiao**[1], **Debra A. Wadford**[2], **Stacia K. Wyman**[4], **Jacek Skarbinski**[1,3,5,6]

**1** Division of Research, Kaiser Permanente Northern California, Oakland, California, United States of America, **2** Viral and Rickettsial Disease Laboratory, Center for Laboratory Sciences, California Department of Public Health, Richmond, California, United States of America, **3** The Permanente Medical Group, Kaiser Permanente Northern California, Oakland, California, United States of America, **4** Innovative Genomics Institute, University of California, Berkeley, California, United States of America, **5** Department of Infectious Diseases, Oakland Medical Center, Kaiser Permanente Northern California, Oakland, California, United States of America, **6** Physician Researcher Program, Kaiser Permanente Northern California, Oakland, California, United States of America

* joshua.r.nugent@kp.org

## Abstract

### Background

The possibility of association between SARS-CoV-2 genomic variation and immune evasion is not known among persons with Omicron variant SARS-CoV-2 infection.

### Methods

In a retrospective cohort, using Poisson regression adjusting for sociodemographic variables and month of infection, we examined associations between individual non-lineage defining mutations and SARS-CoV-2 immunity status, defined as a) no prior recorded infection, b) not vaccinated but with at least one prior recorded infection, c) complete primary series vaccination, and/or d) primary series vaccination and ≥1 booster. We identified all non-synonymous single nucleotide polymorphisms (SNPs), insertions and deletions in SARS-CoV-2 genomes with ≥5% allelic frequency and population frequency of ≥5% and ≤95%. We also examined correlations between the presence of SNPs with each other, with subvariants, and over time.

### Results

Seventy-nine mutations met inclusion criteria. Among 15,566 persons infected with Omicron SARS-CoV-2, 1,825 (12%) were unvaccinated with no prior recorded infection, 360 (2%) were unvaccinated with a recorded prior infection, 13,381 (86%) had a complete primary series vaccination, and 9,172 (58%) had at least one booster. After examining correlation between SNPs, 79 individual non-lineage defining mutations were organized into

**Data availability statement:** Minimal data for this study contain both potentially identifying and sensitive patient health information. According to KPNC IRB standards, these data may only be shared externally if the following criteria/procedures have been met: Anonymized data that support the findings of this study may be made available from the investigative team in the following conditions: (1) agreement to collaborate with the study team on all publications, (2) provision of external funding for administrative and investigator time necessary for this collaboration, (3) demonstration that the external investigative team is qualified and has documented evidence of training for human subjects protections, and (4) agreement to abide by the terms outlined in data use agreements between institutions. Data are available upon request from KPNC IRB via mail (1800 Harrison Street, 10th Floor, Oakland, CA 94612, United States of America), telephone (1-866-241-0690), or email (kpnc.irb@kp.org), for researchers who meet the criteria for access to confidential information.

**Funding:** This study was supported by the Rockefeller Foundation (SKW, JS), National Cancer Institute of the National Institutes of Health under Award Number U01CA260584 as part of the Serological Sciences Network (SeroNet) (JS), the Physician Researcher Program of The Permanente Medical Group Delivery Science and Applied Research Program (JS), The Packard Foundation under Award Number 42937 (SKW), and The Sergey Brin Family Foundation under Award Number 47769 (SKW). CDPH/COVIDNet genomic surveillance work was funded by Centers for Disease Control and Prevention, Epidemiology and Laboratory Capacity for Infectious Diseases, Cooperative Agreement Number 5 NU50CK000539. The funders did not play any role in the study design, data collection and analysis, decision to publish, or preparation of the manuscript.

**Competing interests:** The authors have declared that no competing interests exist.

38 groups. After correction for multiple testing, no individual SNPs or SNP groups were significantly associated with immunity status levels.

## Conclusions

Genomic variation identified within SARS-CoV-2 Omicron specimens was not significantly associated with immunity status, suggesting that contribution of non-lineage defining SNPs to immune evasion is minimal. Larger-scale surveillance of SARS-CoV-2 genomes linked with clinical data can help provide information to inform future vaccine development.

## Introduction

Since its emergence, severe acute respiratory syndrome coronavirus 2 (SARS-CoV-2) has been evolving, with evidence for increased transmissibility and ability to evade existing immune responses elicited by prior infection or COVID-19 vaccination [1,2]. Multiple mutations that increase transmission and immune evasion were identified when the Delta and subsequent Omicron variants (and sub-variants) emerged. However, few in-depth analyses characterizing the genomic variation within a circulating variant and identifying potential mutations associated with immune evasion have been conducted. In addition, during the Omicron wave, individual immunity could be due to prior infection or multiple levels of vaccination, due to the availability of vaccine boosters in this timeframe, and any immune evasion characteristics found among mutations could be tied to a specific type of immunity, derived from, for example, prior infection, immunization dose(s), or both.

To address these data gaps, we conducted a retrospective cohort study of persons with nucleic-acid amplification test (NAAT)-confirmed SARS-CoV-2 infection with the Omicron variant confirmed by whole genome sequencing. Patients were members of a large, integrated health system that had detailed epidemiologic and clinical data on all participants. The study assessed overall SARS-CoV-2 genetic variation over time as well as the relationship between mutations and the individual's source(s) of potential COVID-19 immunity. The key assumption in this study is that after adjusting for sociodemographic characteristics and timeframe, an association between mutation presence and vaccine- or infection-derived immunity could imply greater immune evasion for that mutation.

## Methods

### Setting

Kaiser Permanente Northern California (KPNC) is an integrated health system that serves over 4.5 million members in Northern and Central California and provides comprehensive preventive and curative care in inpatient and outpatient settings across 266 medical offices and 21 hospitals. Members receive most clinical services, including laboratory testing, outpatient, and inpatient care in KPNC facilities. Members have similar sociodemographic characteristics to the population of Northern and Central California [3]. SARS-CoV-2 NAAT for outpatients and hospitalized patients is conducted at a regional laboratory or local hospital laboratories using various platforms and specimens are sent to the California Department of Public Health (CDPH) for whole genome sequencing.

### Study design

We conducted a retrospective cohort study of persons with NAAT-confirmed SARS-CoV-2 infection and whole genome sequencing-confirmed Omicron variant infection and assessed

patterns among non-lineage-defining mutations as well as the association between SARS-CoV-2 genetic variation and immunity source (if any). This immunity was defined by prior reported SARS-CoV-2 infection or vaccination. Epidemiologic and clinical data were obtained from the KPNC Virtual Data Warehouse, a common data model into which standardized data are extracted from clinical and administrative databases including an integrated electronic health record database (Epic, Verona, WI, USA), and from other sources [4,5]. The study was approved by the KPNC and University of California, Berkeley Institutional Review Boards with waivers of the requirement for informed consent. This activity was reviewed by the Centers for Disease Control and Prevention (CDC) and conducted consistent with applicable federal law and CDC policy (see, e.g., 45 C.F.R. part 46.102(l)(2), 21 C.F.R. part 56; 42 U.S.C. §241(d); 5 U.S.C. §552a; 44 U.S.C. §3501 et seq.).

## Inclusion criteria

We included all persons with an incident NAAT-confirmed SARS-CoV-2 infection and whole genome sequencing-confirmed Omicron variant infection from January 1, 2022 to October 31, 2022 who met the following criteria: 1) KPNC membership for at least one year prior to SARS-CoV-2 diagnosis; 2) aged ≥5 years and <90 years; 3) no prior receipt of COVID-19 vaccine or the completion of a primary COVID-19 vaccine series of two doses of BNT162b2 [Pfizer/BioNTech] or m-RNA-1973 [Moderna/National Institutes of Health] or one dose of Ad.26.COV2.s [Johnson & Johnson] more than 14 days prior to index date; 4) incident infection defined as the first NAAT-confirmed SARS-CoV-2 infection during the study period. The index date was defined as the date of NAAT-confirmation of SARS-CoV-2 infection.

## SARS-CoV-2 whole genome sequencing and processing

KPNC is a partner in COVIDNet with CDPH and submits a majority of NAAT-confirmed SARS-CoV-2 specimens to COVIDNet for genomic surveillance; a subset of specimens undergo whole genome sequencing at CDPH or affiliated laboratories based on CDPH priorities, including prioritizing hospitalized cases and a convenience sample of the first specimens received per week (number varies by burden) [6]. COVIDNet consists of a set of (primarily University of California-affiliated) laboratories tapped to assist CDPH because they had surplus sequencing capacity. Illumina sequencing was used by most of the sites, with about 10% of genomes being sequenced by Oxford Nanopore. For this study, we only included Illumina-sequenced genomes because the error rate of Nanopore is too high to accurately identify low-frequency mutations. Varying sets of primers were used by different institutions, but the most common for the Omicron genomes were the ARTICv3 and ARTICv4 primers. The University of California, San Francisco lab used their own custom set of primers with more primers and shorter amplicons. After sequencing, the CDPH Viral and Rickettsial Disease Laboratory links lineage information with patient identifiers and shares the results with KPNC. The raw sequence files are shared with the University of California, Berkeley (UCB); KPNC and UCB collaborate further to link demographic, clinical, and epidemiologic data from KPNC with subsequent sequence analysis data from UCB.

The raw sequencing data were processed through a SARS-CoV-2 analysis pipeline that was modified for this work as follows. Adapter removal and trimming were performed using bbduk (http://sourceforge.net/projects/bbmap/). The reads were then aligned to the Wuhan reference genome (GISAID ID: EPI_ISL_402125, GenBank accession number MN908947) using minimap2 [7] followed by primer trimming using iVar [8]. Samtools was used to create a pileup file from which a consensus sequence was generated. This consensus genome was created with iVar using a minimum depth of 10 reads and majority rule for base calling. We next

used iVar to call variants from the pileup file where we set the sensitivity threshold for calling a mutation to 2%. This process calls mutations for any loci where at least 2% of the reads are non-reference. Using this very low threshold allowed us to capture even very low frequency mutations in the sequencing data. This threshold resulted in many spurious mutations from sequencing and alignment errors and therefore we later set a threshold of 5% when processing mutations. This original list of variants was then annotated with the gene and amino acid change (if there was one) or insertion or deletion.

After calling with iVar using a low-frequency threshold, mutations were compiled into a master frequency matrix listing all mutations that appeared in any genome in rows, and all the genomes that each mutation appeared in columns (https://github.com/staciawyman/ SC2_GenVar). The cell values were frequency of the alternate reads at the location. A mutation had to have an allele frequency of at least 5% to be included to rule out sequencing errors. This matrix was partitioned into variant-specific parts to facilitate analysis. We constructed an Omicron matrix including all individuals that had the Omicron lineage [9] between January 1, 2022 and October 31, 2022 including non-synonymous SNPs, insertions and deletions (indels). Pipeline code is available at https://github.com/staciawyman/SC2_GenVar. For research purposes, data was accessed between January 1, 2023 and June 30, 2024.

### Outcome measure

The primary outcome measure was, among persons with an incident NAAT-confirmed SARS-CoV-2 infection, the presence of individual mutations (nonsynonymous SNPs, insertions and deletions) with ≥5% allelic frequency and ≥5% and ≤95% population frequency, as well as mutation groups created by grouping SNPs with a pairwise Pearson correlation coefficient > 0.7. Additionally, patterns in the prevalence of these mutations were assessed as follows: as noted earlier, pairwise Pearson correlations with other mutations in each infection; presence within Omicron subvariant (BA.1, BA.2, BA.4, BA.5, BCD, BE, BF, Bdot), and prevalence over time.

### Primary exposure: SARS-CoV-2 immunity status

The primary exposure of interest in this analysis was immunity level, defined in mutually exclusive categories: i) no recorded vaccination or recorded prior infection; ii) recorded prior infection but no recorded vaccination; and iii) recorded vaccination (primary series complete and/or more than primary series), regardless of prior infection status.

### Statistical analysis

Adjusted relative prevalence, shown as an adjusted rate ratio (aRR), and 95% confidence intervals (CIs) for the estimates for our exposure and adjustment variables were obtained from modified Poisson regression models with robust standard errors using PROC GENMOD with a log link function in SAS (Version 9.4, Cary, North Carolina); statistical significance was defined as a p-value <0.05. All p-values were adjusted for multiple comparisons to control the false discovery rate at 5%, using the Benjamini-Hochberg procedure [10]. To adjust for confounding, all models included age category, sex, race/ethnicity, Charlson comorbidity index score [11], and index month. Adjusting for these demographic and clinical factors accounted for differential vaccination status of these subgroups, increasing vaccination rates over time, and the inherent uncertainty as to where in a population a new mutation might arise.

### Results

From January 1, 2022 to October 31, 2022, 312,249 SARS-CoV-2-positive specimens were submitted for sequencing, 18,603 (6.0%) underwent sequencing (S1 Fig), and 18,476 (5.9%)

had reported lineages consistent with the Omicron variant. After applying selection criteria, 15,556 persons with SARS-CoV-2 Omicron variant infection were included in the analysis: 9,172 (58%) vaccinated with a booster; 4,209 (26%) vaccinated with a complete primary series [two doses of Pfizer or Moderna vaccine]; and 2,185 (14%) with no vaccination (S1 Fig). The prevalence of Omicron subvariants in our specimens, aggregated to the week level, showed increasing diversity over time (Fig 1).

After applying selection criteria to the sequences, we identified 78 non-lineage defining mutations (all non-synonymous SNPs) in 1,456 SARS-CoV-2 SNPs included in this analysis, shown in S2 Fig. Identified mutations were in the spike gene (n = 44), nucleocapsid gene (n = 8) open reading frame 1a (ORF1a) (n = 19), ORF1b (n = 4), ORF3a (n = 1), ORF6c (n = 1), ORF10 (n = 1). The population frequency within our genomes of individual identified mutations ranged from 5.0% to 95.0% and 46 out of 79 (58.2%) mutations were present in <50% of the overall population. Although we called low-frequency mutations with our pipeline, none of them met the criteria for inclusion in our analysis.

Of the 78 identified SNPs that met our inclusion criteria, many co-occurred on the same specimens. S2 Fig shows the pairwise correlations of SNP presence in our dataset. Given the strong pairwise correlations observed, we formed groups of SNPs that had a pairwise Pearson correlation > .7, reducing the dimension to 33 SNPs/SNP groups, also shown in S2 Fig. These

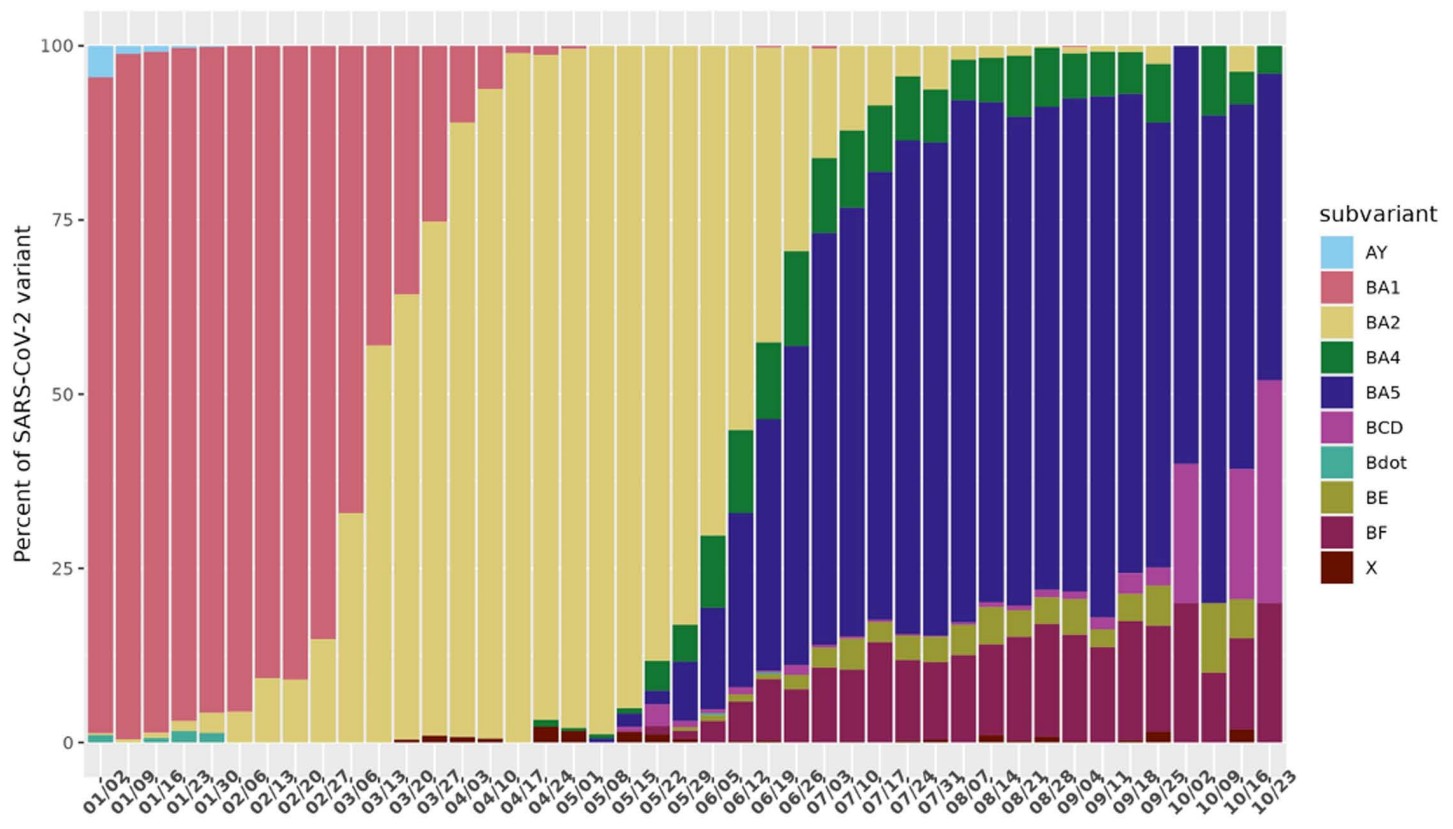

**Fig 1. Specimen subvariant proportions.** SARS-CoV-2-positive specimens tested by Kaiser Permanente Northern California, subvariant proportions by week, January 1, 2022 to October 31, 2022.

mutation groups are largely monophyletic and segregate together across the phylogeny of Omicron genomes (S3 Fig).

The demographics of persons with SARS-CoV-2 infection included in this analysis, stratified by vaccination and prior infection status, are shown in Table 1, and the number of total reported infections and those sequenced over time is shown in S4 Fig. Significant differences were found between groups, thus supporting their inclusion in our covariate adjustment set. Figure 2 details some of the relationships between SNP groups, time, and subvariant among the specimens in our data set. While certain mutations/groups appear essentially only in a single subvariant (for example, 99.1% of S:H69L mutations appear in specimens classified as BA.1 in Figure 2c), the majority of mutations/groups appear in many subvariants, suggesting that they are not simply proxy measures of subvariant. Further, the presence of mutations/groups do not follow clear temporal patterns (Figure 2b). While some mutations such as Group 12 arise, become plentiful, then wane, others disappear and re-appear over time without a clear pattern (for example, S:D1259H). Smoothed over time, grouped SNP prevalence by month is shown in S5 Fig.

Using modified Poisson regression models with robust standard errors, we found several of our 33 SNP groupings (Table 2) and/or 78 individual mutations (S1 Table) that were significantly associated with immunity status; however, after correcting p-values to adjust for multiple comparisons, none of the associations were statistically significant. These findings were not meaningfully changed when people with a booster vaccine were/were not aggregated with the complete primary series group (S2 Table, S3 Table). The SNPs/groups with the largest absolute associations with vaccination (Table 2) included some that were strongly associated with a single variant (for example, ORF10:L37F, which occurred almost exclusively in BA.5 subvariants) as well as some that frequently appeared in multiple variants (for example, ORF-1b:N498I and group 45).

When looking at the evolutionary relationships with the genomes colored by vaccination status (S6 Fig), vaccination status did not show a correlation with a clade/subvariant or distance from the root (number of mutations with respect to the reference SARS-CoV-2 genome).

## Discussion

Characterizing the continuing evolution of SARS-CoV-2, along with active surveillance of infections, is critical to targeting public health measures and vaccine development. Large-scale global whole genome sequencing efforts have contributed substantially to our understanding of the genetic variation of SARS-CoV-2 and its associations with transmissibility and immune evasion. Our current study adds to this evidence base by assessing within-variant (Omicron) variation in the genome beyond the already-identified subvariants such as BA.1 and BA.2 using a unique whole genome surveillance system linking genomic data with detailed data on demographic, clinical and epidemiologic characteristics in a well-defined population.

Understanding the potential for immune evasion based on mutations beyond the spike protein is critical for vaccine development and understanding the mechanisms of the human immune response. In this detailed analysis of genomic variation within the Omicron variant over a 10-month period in California, we identified only 78 mutations that met our inclusion criteria among 15,566 genomes. These mutations occurred across the entire genome, not just in the spike protein. Spike protein mutations previously implicated with immune evasion were excluded based on population frequency. We encourage future research to examine the entire viral genome for potential immune evasion or increased replication or infectivity.

Our results showed that there appear to be some mutations that are tightly grouped with certain subvariants, while a number of other mutations appear across multiple

**Table 1. Cohort characteristics.**

| Characteristic | All n (%) | Not vaccinated, no prior reported infection n (%) | Not vaccinated, prior reported infection n (%) | Vaccinated n (%) | p-value[1] |
|---|---|---|---|---|---|
| **Total** | 15,566 (100%) | 1,825 (11.7%) | 360 (2.3%) | 13,381 (86.0%) | |
| **Median age in years (Interquartile range)** | 43.0 (30.0, 57.0) | 32.0 (13.0, 49.0) | 32.0 (19.0, 43.0) | 44.0 (32.0, 58.0) | <0.0001 |
| **Age in years** | | | | | <0.0001 |
| 5-19 | 2,019 (13.0%) | 594 (32.5%) | 95 (26.4%) | 1,330 (9.9%) | |
| 20-29 | 1,776 (11.4%) | 219 (12.0%) | 57 (15.8%) | 1,500 (11.2%) | |
| 30-39 | 3,054 (19.6%) | 323 (17.7%) | 97 (26.9%) | 2,634 (19.7%) | |
| 40-49 | 2,859 (18.4%) | 242 (13.3%) | 53 (14.7%) | 2,564 (19.2%) | |
| 50-59 | 2,572 (16.5%) | 189 (10.4%) | 35 (9.7%) | 2,348 (17.5%) | |
| 60-69 | 1,879 (12.1%) | 135 (7.4%) | 13 (3.6%) | 1,731 (12.9%) | |
| 70-79 | 1,023 (6.6%) | 88 (4.8%) | 8 (2.2%) | 927 (6.9%) | |
| 80-89 | 384 (2.5%) | 35 (1.9%) | 2 (0.6%) | 347 (2.6%) | |
| **Sex** | | | | | 0.0003 |
| Female | 8,665 (55.7%) | 929 (50.9%) | 204 (56.7%) | 7,532 (56.3%) | |
| Male | 6,901 (44.3%) | 896 (49.1%) | 156 (43.3%) | 5,849 (43.7%) | |
| **Race/ethnicity** | | | | | <0.0001 |
| LatinX/Hispanic ethnicity | 4,303 (27.6%) | 591 (32.4%) | 158 (43.9%) | 3,554 (26.6%) | |
| Black/African descent | 1,368 (8.8%) | 267 (14.6%) | 45 (12.5%) | 1,056 (7.9%) | |
| Asian descent | 4,090 (26.3%) | 211 (11.6%) | 38 (10.6%) | 3,841 (28.7%) | |
| White/European or middle eastern descent | 4,788 (30.8%) | 603 (33.0%) | 100 (27.8%) | 4,085 (30.5%) | |
| Other/unknown | 1,017 (6.5%) | 153 (8.4%) | 19 (5.3%) | 845 (6.3%) | |
| **Charlson comorbidity index score** | | | | | 0.44 |
| Score 0 | 10,082 (64.8%) | 1,218 (66.7%) | 271 (75.3%) | 8,593 (64.2%) | |
| Score 1 | 2,278 (14.6%) | 225 (12.3%) | 48 (13.3%) | 2,005 (15.0%) | |
| Score 2 | 881 (5.7%) | 60 (3.3%) | 4 (1.1%) | 817 (6.1%) | |
| Score 3 + | 1,141 (7.3%) | 118 (6.5%) | 11 (3.1%) | 1,012 (7.6%) | |
| Missing (no visits in prior year) | 1,184 (7.6%) | 204 (11.2%) | 26 (7.2%) | 954 (7.1%) | |
| **COVID-19 vaccination** | | | | | |
| None | 2,185 (14.0%) | 1,825 (100.0%) | 360 (100.0%) | | |
| Complete primary series | | | | | |
| Johnson & Johnson | 803 (5.2%) | | | 803 (6.0%) | |
| Moderna | 4,359 (28.0%) | | | 4,359 (32.6%) | |
| Pfizer | 8,219 (52.8%) | | | 8,219 (61.4%) | |
| **Month of SARS-CoV-2 infection** | | | | | 0.007 |
| January 2022 | 3,120 (20.0%) | 663 (36.3%) | 65 (18.1%) | 2,392 (17.9%) | |
| February 2022 | 1,280 (8.2%) | 250 (13.7%) | 26 (7.2%) | 1,004 (7.5%) | |
| March 2022 | 791 (5.1%) | 92 (5.0%) | 13 (3.6%) | 686 (5.1%) | |
| April 2022 | 1,090 (7.0%) | 88 (4.8%) | 10 (2.8%) | 992 (7.4%) | |
| May 2022 | 1,210 (7.8%) | 95 (5.2%) | 21 (5.8%) | 1,094 (8.2%) | |
| June 2022 | 1,837 (11.8%) | 152 (8.3%) | 53 (14.7%) | 1,632 (12.2%) | |
| July 2022 | 2,770 (17.8%) | 212 (11.6%) | 75 (20.8%) | 2,483 (18.6%) | |
| August 2022 | 2,263 (14.5%) | 182 (10.0%) | 62 (17.2%) | 2,019 (15.1%) | |
| September 2022 | 1,070 (6.9%) | 83 (4.5%) | 32 (8.9%) | 955 (7.1%) | |
| October 2022 | 135 (0.9%) | 8 (0.4%) | 3 (0.8%) | 124 (0.9%) | |
| **Months from last vaccine dose to SARS-CoV-2 infection** | | | | | |
| Mean (standard deviation) | 6.5 (3.7) | | | 6.5 (3.7) | |
| Median (25th percentile, 75th percentile) | 6.3 (3.7, 8.6) | | | 6.3 (3.7, 8.6) | |

*(Continued)*

**Table 1.** (Continued)

| Characteristic | All n (%) | Not vaccinated, no prior reported infection n (%) | Not vaccinated, prior reported infection n (%) | Vaccinated n (%) | p-value[1] |
|---|---|---|---|---|---|
| **Immune compromising conditions[2]** | | | | | |
| Any immune compromising condition | 1,092 (7.0%) | 92 (5.0%) | 16 (4.4%) | 984 (7.4%) | 0.0009 |
| **Severe clinical outcomes** | | | | | |
| Any hospitalization | 483 (3.1%) | 164 (9.0%) | 5 (1.4%) | 314 (2.3%) | <0.0001 |

Demographics of persons included in this analysis with SARS-CoV-2 infection with Omicron variant in Northern California stratified by vaccination/infection status, January 1, 2022 - October 31, 2022.

[1]Wilcoxon rank sum test; Pearson's Chi-squared test.

[2]Immune compromising conditions: HIV, organ transplant, cancer, rheumatologic/inflammatory conditions, and other immune deficiencies.

subvariants. While we did not find evidence for immune evasion among our identified mutations/groups after correction for multiple testing, it is possible that we are under-powered to detect small effects over a short time period [12]. Many of the SNPs with the highest adjusted risk ratios appeared in a small number of specimens (S1 Table). While one would expect more variability from the smallest table cells if we were simply observing statistical noise, it is also possible that a larger sample could confirm these stronger adjusted risk ratios.

Our analysis has several limitations. First, not all possible demographic and clinical variables were included in our models, and if those variables are associated with immunity status and SNP presence, unmeasured confounding could bias our analysis. Second, there is a risk of misclassification of immunity status; though the KPNC surveillance system for vaccination is quite robust, prior infections (say, diagnosed by a home test) may have gone unreported. Third, we focused on a set of specimens from Northern and Central California, limiting the generalizability of our findings to this context. Finally, there may be hidden bias as to which specimens were successfully sequenced based on the nucleic acid viral load in the specimen (e.g., specimens with lower viral loads might not be successfully sequenced and thus would have been excluded from this analysis). Fourth, although our surveillance system has a very robust system for assessing individual-level vaccination status, there is a small risk of misclassification of persons as having incorrect immunity status.

As part of the global effort to conduct near real-time genomic surveillance for SARS-CoV-2, there has been tremendous progress in developing national genomic surveillance systems with nearly real-time public reporting of circulating SARS-CoV-2 variants and uploading of consensus sequence data to publicly available genomic databases [13–16]. However, these efforts can be improved. First, enhancing methods to define consensus sequences to also assess for minor variation with <50% allelic frequency would better define the true genomic variation of circulating SARS-CoV-2, including capturing within-host genetic diversity [17,18]. Second, current SARS-CoV-2 genomic surveillance systems in the United States have limited metadata on human demographics, clinical factors, epidemiologic factors, and immunity status (e.g., COVID-19 vaccination and documented prior infection), particularly when compared to the United Kingdom [19]. This limits the ability to correlate noted patterns in SARS-CoV-2 genetic variation and real-world clinical and epidemiologic outcomes in the United States. We present one potential analysis that uses this data linkage to better understand how SARS-CoV-2 genetic variation might be associated with human disease, in this case evidence for immune evasion associated with specific mutation clusters. Improving the clinical metadata associated with SARS-CoV-2 genomic data should be a priority as we continue to improve and expand genomic surveillance. We have demonstrated this capability through

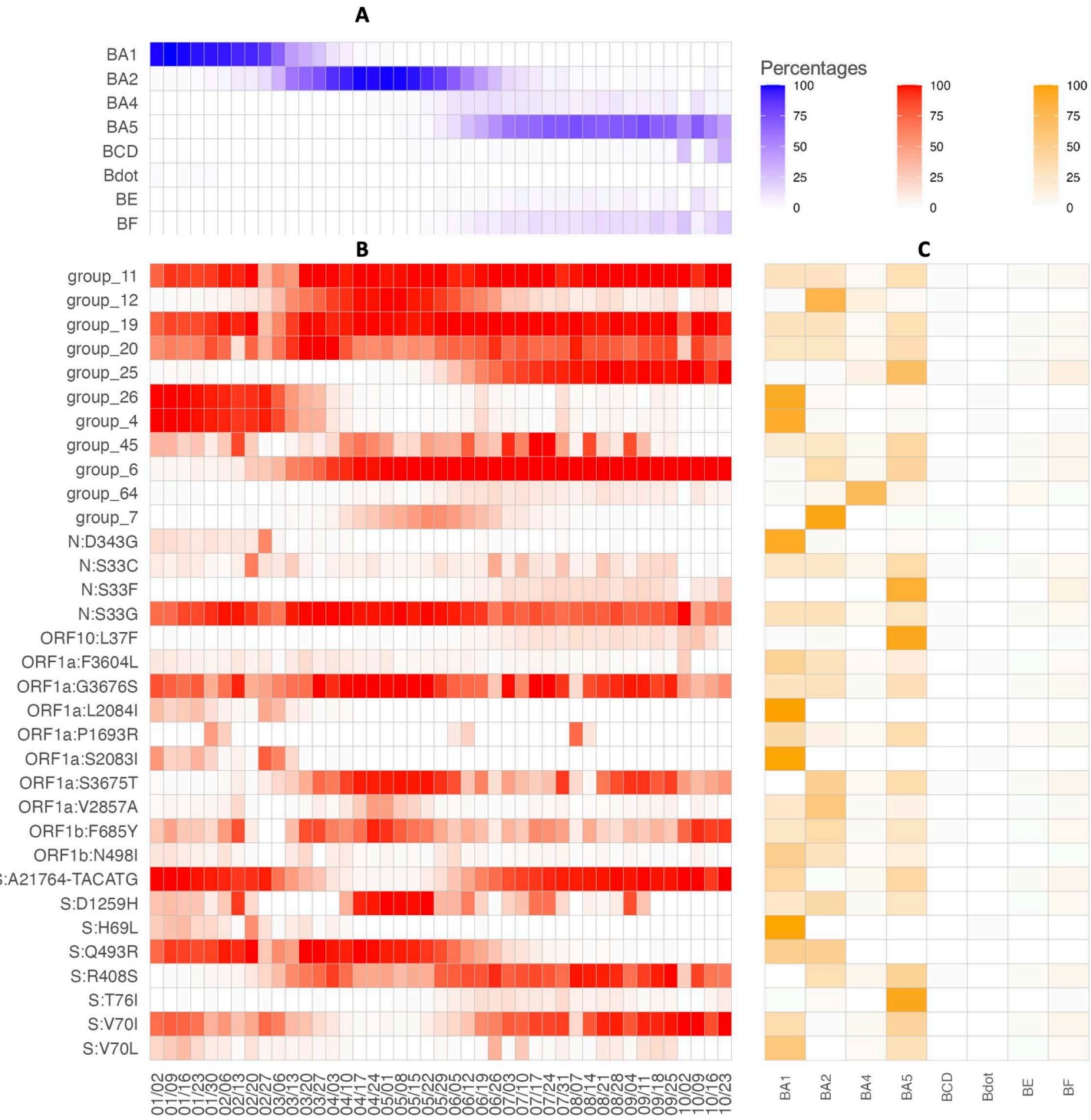

**Fig 2. Relationships between SNP group prevalence, subvariant, and time.** SNP, group, and sublineage appearance, January 1, 2022 to October 31, 2022: (a) Proportion of specimens in each sublineage by week; (b) Proportion of specimens with a given SNP/group by week; (c) Proportion of sublineage by SNP/group (rows sum to 100%).

**Table 2. Associations between SNP group presence and immunity status.**

| SNP/group | Not vaccinated, no recorded prior infection | Not vaccinated, recorded prior infection | | | Vaccinated, regardless of prior infection status | | |
|---|---|---|---|---|---|---|---|
| | | aRR (95% CI) | p (raw) | p (FDR) | aRR (95% CI) | p (raw) | p (FDR) |
| group64 | Reference | 1.26 (0.82- 1.93) | 0.2845 | 0.97743 | 1.22 (0.96- 1.56) | 0.1097 | 0.72385 |
| ORF10:L37F | Reference | 1.00 (0.61- 1.62) | 0.9849 | 0.98491 | 1.18 (0.91- 1.53) | 0.2210 | 0.86611 |
| ORF1b:N498I | Reference | 0.96 (0.57- 1.63) | 0.8889 | 0.97743 | 1.13 (0.92- 1.40) | 0.2404 | 0.86611 |
| group45 | Reference | 1.13 (0.96- 1.34) | 0.1504 | 0.97743 | 1.10 (1.01- 1.19) | 0.0282 | 0.46532 |
| ORF1a:V2857A | Reference | 1.38 (0.84- 2.29) | 0.2047 | 0.97743 | 1.09 (0.86- 1.39) | 0.4703 | 0.90946 |
| S:D1259H | Reference | 1.06 (0.87- 1.29) | 0.5509 | 0.97743 | 1.08 (0.99- 1.19) | 0.0834 | 0.68786 |
| S:T76I | Reference | 1.25 (0.80- 1.95) | 0.3219 | 0.97743 | 1.07 (0.83- 1.40) | 0.5906 | 0.91512 |
| N:S33C | Reference | 1.07 (0.78- 1.45) | 0.6751 | 0.97743 | 1.06 (0.92- 1.24) | 0.4194 | 0.89674 |
| group25 | Reference | 1.04 (0.88- 1.23) | 0.6343 | 0.97743 | 1.04 (0.95- 1.14) | 0.4105 | 0.89674 |
| S:V70L | Reference | 0.89 (0.62- 1.29) | 0.5376 | 0.97743 | 1.04 (0.90- 1.20) | 0.6297 | 0.91512 |
| group7 | Reference | 0.95 (0.66- 1.36) | 0.7830 | 0.97743 | 1.04 (0.87- 1.24) | 0.6892 | 0.91512 |
| ORF1a:F3604L | Reference | 0.98 (0.59- 1.64) | 0.9420 | 0.97743 | 1.04 (0.84- 1.28) | 0.7322 | 0.91512 |
| group19 | Reference | 1.02 (0.91- 1.15) | 0.6952 | 0.97743 | 1.03 (0.98- 1.09) | 0.2357 | 0.86611 |
| N:S33G | Reference | 0.97 (0.85- 1.10) | 0.6454 | 0.97743 | 1.03 (0.98- 1.09) | 0.2335 | 0.86611 |
| ORF1a:G3676S | Reference | 1.02 (0.90- 1.16) | 0.7179 | 0.97743 | 1.03 (0.97- 1.09) | 0.4073 | 0.89674 |
| ORF1b:F685Y | Reference | 1.01 (0.84- 1.21) | 0.9202 | 0.97743 | 1.03 (0.95- 1.12) | 0.4348 | 0.89674 |
| S:H69L | Reference | 0.61 (0.36- 1.03) | 0.0638 | 0.97743 | 1.03 (0.88- 1.19) | 0.7379 | 0.91512 |
| group11 | Reference | 0.99 (0.88- 1.11) | 0.8721 | 0.97743 | 1.02 (0.97- 1.08) | 0.4105 | 0.89674 |
| S:Q493R | Reference | 0.92 (0.78- 1.10) | 0.3603 | 0.97743 | 1.02 (0.95- 1.09) | 0.6063 | 0.91512 |
| group12 | Reference | 0.92 (0.74- 1.15) | 0.4770 | 0.97743 | 1.02 (0.92- 1.13) | 0.6863 | 0.91512 |
| S:V70I | Reference | 1.03 (0.89- 1.19) | 0.6590 | 0.97743 | 1.01 (0.95- 1.08) | 0.6802 | 0.91512 |
| S:A21764-TACATG | Reference | 1.04 (0.92- 1.18) | 0.5241 | 0.97743 | 1.01 (0.95- 1.07) | 0.8278 | 0.93114 |
| group6 | Reference | 0.99 (0.86- 1.13) | 0.8384 | 0.97743 | 1.01 (0.94- 1.08) | 0.7811 | 0.92055 |
| S:R408S | Reference | 1.01 (0.87- 1.18) | 0.8820 | 0.97743 | 1.01 (0.93- 1.09) | 0.8725 | 0.93114 |
| ORF1a:S3675T | Reference | 0.96 (0.80- 1.16) | 0.7049 | 0.97743 | 1.01 (0.92- 1.10) | 0.9048 | 0.93114 |
| group20 | Reference | 1.00 (0.88- 1.15) | 0.9478 | 0.97743 | 1.00 (0.94- 1.07) | 0.9311 | 0.93114 |
| group26 | Reference | 1.06 (0.88- 1.29) | 0.5288 | 0.97743 | 0.99 (0.92- 1.06) | 0.7487 | 0.91512 |
| ORF1a:P1693R | Reference | 1.06 (0.68- 1.63) | 0.8043 | 0.97743 | 0.99 (0.81- 1.21) | 0.9139 | 0.93114 |
| group4 | Reference | 1.06 (0.88- 1.28) | 0.5443 | 0.97743 | 0.98 (0.91- 1.05) | 0.4961 | 0.90946 |
| N:D343G | Reference | 0.59 (0.31- 1.11) | 0.1019 | 0.97743 | 0.90 (0.75- 1.08) | 0.2628 | 0.86611 |
| N:S33F | Reference | 1.34 (0.94- 1.92) | 0.1096 | 0.97743 | 0.89 (0.71- 1.11) | 0.2887 | 0.86611 |
| ORF1a:S2083I | Reference | 1.06 (0.74- 1.53) | 0.7369 | 0.97743 | 0.87 (0.76- 1.00) | 0.0431 | 0.47401 |
| ORF1a:L2084I | Reference | 0.90 (0.58- 1.39) | 0.6213 | 0.97743 | 0.84 (0.72- 0.97) | 0.0177 | 0.46532 |

Association between mutation groups[1] and vaccine status (pooling over complete and boosted) among persons included in this analysis with SARS-CoV-2 infection with Omicron variant in Northern California, January 1, 2022 - October 31, 2022 (N = 15,566)[2]

[1]Mutations with Pearson pairwise correlation > 0.70 combined into groups for analysis:

group 4: M:D3G, ORF1a:I3758V, ORF1a:K856R, S:A67V, S:G496S, S:N211I, S:N856K, S:R346K,

S:S371P, S:T22204 + GAGCCAGAA, S:T547K, S:T95I.

group 6: N:S413R, ORF1a:F3677L, ORF1a:G1307S, ORF1a:L3027F, ORF1a:S135R,

ORF1a:T3090I, ORF1a:T842I, ORF1b:R1315C, ORF1b:T2163I, ORF3a:T223I,

S:D405N, S:G142D, S:L24S, S:T19I, S:T376A, S:V213G.

group 7: S:L452Q, S:S704L.

group 11: S:E484A, S:N501Y, S:Q498R, S:S477N, S:T478K, S:Y505H.

group 12: ORF1a:L3201F, ORF6:D61L.

group 19: S:S371F, S:S373P, S:S375F.

*(Continued)*

**Table 2.** (Continued)

group 20: S:K417N, S:N440K.

group 25: M:D3N, S:F486V, S:L452R.

group 26: ORF1a:A2710T, ORF1a:L3674F, S:G21986-GTGTTTATT, S:G22193-ATT, S:G446S, S:L981F.

group 45: S:A27S, S:C1254 *

group 64: N:P151S, ORF1a:F143L.

[2]Adjusted risk ratios derived from Poisson regression models to assess the association of immunity status with each mutation group adjusting for age, sex, race/ethnicity, Charlson comorbidity index score, and month of SARS-CoV-2 infection.

partnership and collaboration between a large healthcare system and a state public health department, namely KPNC and CDPH/COVIDNet.

In summary, we demonstrate that linking SARS-CoV-2 genomic sequence data with detailed clinical data revealed no significant associations between genetic variation among Omicron SARS-CoV-2 mutations and immunity status. Further research using larger sample sets that link SARS-CoV-2 genetic data with clinical data is warranted for the potential to reveal whether SARS-CoV-2 non-lineage defining genetic variation might contribute to the fitness of SARS-CoV-2 by either immune evasion or other mechanism.

## Supporting information

**S1 Fig. Cohort.** Cohort eligibility flowchart with prior recorded infection and vaccination status.
(TIF)

**S2 Fig. SNP correlation.** Pearson correlations of SNP presence in specimens, before and after clustering.
(TIF)

**S3 Fig. Phylogenies.** Phylogenies of 15,045 Omicron genomes colored by group. Mutations in group 52 (top) represent a clade defined by the BA.2.12.1 subvariant (in blue). Mutations in group 4 (bottom) fall into a subclade of BA.4 (in yellow).
(TIF)

**S4 Fig. Total and sequenced SARS-CoV-2 infections.** Persons with recorded SARS-CoV-2 infection at KPNC by week from January 1, 2022 to October 31, 2022.
(TIF)

**S5 Fig. SNP prevalence.** Proportion of specimens with a given SNP or group, by month, January–October 2022.
(TIF)

**S6 Fig. Phylogenies by vaccination status.** Phylogeny of 15,045 Omicron genomes sampled between 1/1/2022 and 9/25/22. Branch lengths represent divergence from Wuhan reference genome and nodes are colored by vaccination status. Vaccinated category includes primary series and primary plus booster.
(TIF)

**S1 Table. Individual SNP associations with vaccination status.** Association between SNP presence and vaccine status (pooling over complete and boosted) among persons included in this analysis with SARS-CoV-2 infection with Omicron variant in Northern California, January 1, 2022 - October 31, 2022 (N = 15,566).
(TIF)

**S2 Table. Individual SNP associations with vaccination substatus.** Association between SNP presence and vaccine status (separating out complete and boosted) among persons included in this analysis with SARS-CoV-2 infection with Omicron variant in Northern California, January 1, 2022 – October 31, 2022 (N = 15,566).
(TIF)

**S3 Table. SNP and group associations with vaccination substatus.** Association between mutation groups and vaccine status (separating complete and boosted) among persons included in this analysis with SARS-CoV-2 infection with Omicron variant in Northern California, January 2022–October 2022 (N = 15,566). Adjusted risk ratios derived from Poisson regression models to assess the association of immunity status with each mutation group adjusting for age, sex, race/ethnicity, Charlson comorbidity index score, and month of SARS-CoV-2 infection.
(TIF)

## Acknowledgments

The authors are grateful to all Kaiser Permanente members without whom this study would not have been possible. In addition, we wish to acknowledge our partnership with the California Department of Public Health and participation in the California SARS-CoV-2 Whole Genome Sequencing Initiative (COVIDNet). We thank the VRDL and COVIDNet Teams: Summer Adams, Allison Bailey, Matt Bacinskas, Nikki Baumrind, Elizabeth Baylis, John Bell, Ricardo Berumen III, Yocelyn Cruz, Mojgan Deldari, Alex Espinosa, Sabrina Gilliam, Madeline Glenn, Bianca Gonzaga, Melanie Greengard, Jill Hacker, Kim Hansard, Monica Haw, Thalia Huynh, Chantha Kath, Ruth Lopez, Sharon Messenger, Alexa Quintana, Chris Preas, Clarence Reyes, Maria Salas, Hilary Tamnanchit, Serena Ting, Kathy Jacobson, MD, and Carol Glaser, MD.

The content is solely the responsibility of the authors and does not represent the official views or opinions of the National Institutes of Health, Kaiser Permanente, California Department of Public Health or the California Health and Human Services Agency.

Use of trade names and commercial sources is for identification only and does not imply endorsement by the California Department of Public Health or the California Health and Human Services Agency.

## Author contributions

**Conceptualization:** Mariah S. Wood, Liyan Liu, Jeffrey M. Schapiro, Laura B. Amsden, Crystal A. Hsiao, Stacia K. Wyman, Jacek Skarbinski.

**Formal analysis:** Joshua R Nugent, Mariah S. Wood, Liyan Liu, Teal Bullick, Stacia K. Wyman, Jacek Skarbinski.

**Investigation:** Joshua R Nugent.

**Methodology:** Joshua R Nugent, Mariah S. Wood, Liyan Liu, Jeffrey M. Schapiro, Laura B. Amsden, Crystal A. Hsiao, Stacia K. Wyman, Jacek Skarbinski.

**Supervision:** Debra A. Wadford, Jacek Skarbinski.

**Visualization:** Joshua R Nugent.

**Writing – original draft:** Joshua R Nugent, Stacia K. Wyman, Jacek Skarbinski.

**Writing – review & editing:** Joshua R Nugent, Mariah S. Wood, Liyan Liu, Teal Bullick, Jeffrey M. Schapiro, Phacharee Arunleung, Gautham Gautham, Shiffen Getabecha, Christina Morales, Laura B. Amsden, Crystal A. Hsiao, Debra A. Wadford, Stacia K. Wyman, Jacek Skarbinski.

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
