## [Decision Letter · Decision Letter 0]

5 Dec 2024

PONE-D-24-39315SARS-CoV-2 Omicron subvariant genomic variation associations with immune evasion in Northern California: A retrospective cohort studyPLOS ONE

Dear Dr. Nugent,

Thank you for submitting your manuscript to PLOS ONE. After careful consideration, we feel that it has merit but does not fully meet PLOS ONE’s publication criteria as it currently stands. Therefore, we invite you to submit a revised version of the manuscript that addresses the points raised during the review process.

We look forward to receiving your revised manuscript.

Kind regards,

Victor C Huber

Academic Editor

PLOS ONE

2. In the online submission form, you indicated that [Data cannot be shared publicly due to HIPPA laws. De-identified data may be supplied upon request.].

4. Please include your tables as part of your main manuscript and remove the individual files. Please note that supplementary tables (should remain/ be uploaded) as separate "supporting information" files

Reviewers' comments:

Reviewer's Responses to Questions

**Comments to the Author**

1. Is the manuscript technically sound, and do the data support the conclusions?

Reviewer #1: Yes

Reviewer #2: Yes

2. Has the statistical analysis been performed appropriately and rigorously? 

Reviewer #1: Yes

Reviewer #2: I Don't Know

3. Have the authors made all data underlying the findings in their manuscript fully available?

Reviewer #1: Yes

Reviewer #2: Yes

4. Is the manuscript presented in an intelligible fashion and written in standard English?

Reviewer #1: Yes

Reviewer #2: Yes

5. Review Comments to the Author

Reviewer #1: The authors have presented a detailed and thoroughly investigated manuscript, exploring the possibility of association between SARS-CoV-2 genomic variation and immune evasion, not observed in patients infected with the Omicron variant strain. The manuscript has provided all the data as well as the statistical analysis, in a way which is easy to read and comprehend.

Reviewer #2: Given the small sample size, I feel the work presented here was thorough enough for the conclusion made. A minor comment would be some of the demographic data may not apply to their study population and can be deleted to reduce confusion, eg, sequences from those who got a Johnson vaccine were not analysed by the study, from what I read.

6. PLOS authors have the option to publish the peer review history of their article (what does this mean? ). If published, this will include your full peer review and any attached files.

**Do you want your identity to be public for this peer review?** For information about this choice, including consent withdrawal, please see our Privacy Policy .

Reviewer #1: No

Reviewer #2: No

---

## [Editor Report · Decision Letter 1]

29 Jan 2025

SARS-CoV-2 Omicron subvariant genomic variation associations with immune evasion in Northern California: A retrospective cohort study

PONE-D-24-39315R1

Dear Dr. Nugent,

We’re pleased to inform you that your manuscript has been judged scientifically suitable for publication and will be formally accepted for publication once it meets all outstanding technical requirements.

Kind regards,

Victor C Huber

Academic Editor

PLOS ONE
---

## [Editor Report · Acceptance letter]

PONE-D-24-39315R1

PLOS ONE

Dear Dr. Nugent,

I'm pleased to inform you that your manuscript has been deemed suitable for publication in PLOS ONE. Congratulations! Your manuscript is now being handed over to our production team.

Kind regards,

on behalf of

Dr. Victor C Huber

Academic Editor

PLOS ONE